# Nanocomposite Membranes Based on Fluoropolymers for Electrochemical Energy Sources

**DOI:** 10.3390/membranes12100935

**Published:** 2022-09-27

**Authors:** Irina Falina, Natalia Kononenko, Sergey Timofeev, Michail Rybalko, Ksenia Demidenko

**Affiliations:** 1Physical Chemistry Department, Kuban State University, 350040 Krasnodar, Russia; 2JSC Plastpolymer, 195197 Saint-Petersburg, Russia

**Keywords:** perfluorinated membrane, inert fluoropolymer, conductivity, diffusion permeability, current-voltage curve, stress-strain curve, extended tree-wire model, redox flow battery, direct methanol fuel cell, solution casting

## Abstract

The physicochemical and transport properties (ion-exchange capacity, water content, diffusion permeability, conductivity, and current-voltage characteristic) of a series of perfluorinated membranes with an inert fluoropolymer content from 0 to 40%, obtained by polymer solution casting, were studied. Based on the analysis of the parameters of the extended three-wire model, the effect of an inert component on the path of electric current flow in the membrane and its selectivity were estimated. The mechanical characteristics of the membranes, such as Young’s modulus, yield strength, tensile strength, and relative elongation, were determined from the dynamometric curves. The optimal amount of the inert polymer in the perfluorinated membrane was found to be 20%, which does not significantly affect its structure and electrotransport properties but increases the elasticity of the obtained samples. Therefore, the perfluorinated membrane with 20% of inert fluoropolymer is promising for its application in redox flow batteries and direct methanol fuel cells.

## 1. Introduction

The expansion of alternative energy applications is closely related to the development of redox flow batteries (RFB) with aqueous and non-aqueous electrolytes, which are widely recognized as being easy for scale up and suitable for large-scale energy storage applications (10 kW–10 MW). The total energy output depends on the volume of the reservoirs [1,2,3]. There are several requirements for RFB membranes, including high conductivity, thermal and chemical stability, mechanical strength, low permeability to electrochemically active components to achieve high Coulomb, and energy efficiency [1,2,3,4,5,6]. Similar demands are placed on proton exchange membranes for direct methanol fuel cells (DMFC) [7,8]. The last requirement could be satisfied by different modification techniques. For example, they are the formation the layer on the membrane surface or introduction of a modifier into the membrane volume, similarly charged with the polymer matrix [9,10]. Therefore, the concentration of ionogenic groups increases and the transfer of co-ions through the membrane decreases. However, this method does not reduce the flow of non-electrolytes through the membrane, which is important for redox batteries, where neutral molecules, such as bromine or organic molecules, act as electroactive particles. The other approach is the introduction the oppositely charged modifier [11,12]. However, the selectivity of the membrane could both improve [12] and reduce [13]. The last effect is undesirable for the systems containing both charged and neutral species.

Another method is combining the charged polymer solution with a modifying agent and subsequent membrane casting [7,9,11,14]. Thus, the incorporation of inert polymers, such as PVDF, PBI, poly (arylene ether ketone), etc. [8,11,15,16,17], into the membranes reduces the membrane swelling and, as a result, the crossover of uncharged particles and the transfer of co-ions. Depending on the production method, materials with a random or regular distribution of the inert component can be produced. In some cases, the reinforcing components are commercial porous materials [18,19] or carcasses obtained by electrospun method [15,20]. Afterwards, the voids are filled with polyelectrolyte by casting the polymer solution with its subsequent distribution over the volume of the material. The casting of a membrane from a solution containing both an inert component and a polyelectrolyte is also used [11,15,18]. The advantage of this method compared to the previous ones is the formation of a pseudo-homogeneous film, which has a high degree of homogeneity in the distribution of charged groups over the membrane volume, which makes it possible to maintain high selectivity values.

The membrane modifying permits researchers to solve the problem of improving some characteristics of the material while slightly reducing others. The introduction of any modifier affects the key characteristics of membranes for redox flow batteries, such as ion-exchange capacity, water content, osmotic permeability, conductivity, diffusion permeability to electrolytes and non-electrolytes and mechanical strength, as well as electroosmotic permeability, since according to [21], non-electrolyte molecules could be involved in electroosmotic transfer.

During RFB operation, the membrane is subjected to various influences, for example, leading to the degradation of polymer chains and the appearance of cavities and cracks in their structure. So, during operation of both batteries and hydrogen fuel cells, the membrane is subjected to highly active oxidants [22]. In this regard, perfluorinated cation exchange membranes with sulfonic acid groups are especially promising objects for completing redox batteries. The reinforcement of perfluorinated membranes with nets of tetrafluoroethylene leads to an increase in their strength, but at the same time, the level of heterogeneity of these materials increases significantly and, as a result, their selectivity decreases [23]. The addition of an inert fluoropolymer during the manufacture of membranes can solve this problem, and this technique has been used in the preparation of Nafion membranes in [24]. However, the question of the optimal ratio of fluoropolymers in the membrane composition remains open, since the introduction of an inert component leads to a decrease in its electrical conductivity. For perfluorinated MF-4SK membranes produced in Russia, such studies have not been previously carried out. In this regard, the aim of this work was to obtain nanocomposite membranes based on perfluorosulfonic acid and inert fluoropolymers with a variable content of an inert component and to study their transport and mechanical properties.

## 2. Materials and Methods

### 2.1. Materials

LF-4SK solution of perfluorosulfonic acid (9 wt.%) in DMF (JSC Plastpolymer, St.-Petersburg, Russia), poly(vinylidene fluoride co-hexafluoropropylene) F-26.

### 2.2. Membrane Preparation

The membrane samples with different content of inert polymer were obtained by mixing solutions of F-26 in DMF and LF-4SK with different ratios. A solution of F-26 in DMF was prepared by ultrasonic dissolution of F-26 powder in DMF. Mixed LF-4SK and F-26 solution was also mixed under the action of ultrasound. Afterwards, the mixed solution was placed into the mold and dried at 60 °C for 2 h and at room temperature for 24 h. The membrane was then extracted from the mold and immersed in 2 M solution of NaCl. The series of the membranes with varied content (ω) of F-26 in the range 0–40 wt.% was prepared. The obtained membranes were marked MX, where X was the weight fraction of F-26 in dry membrane. The main physicochemical characteristics (thickness l, ion-exchange capacity (IEC) Q), water content W, specific water content nm) were obtained according to standard methods described in detail in [25] and are presented in Table 1.

### 2.3. Transport Properties

The resistance of the samples was determined by the two-electrode mercury-contact method as an active part of the impedance of the cell with membrane. The conductivity of the membrane κ_m_ was calculated by the formula
(1)$$\kappa_{m}=\frac{l}{R_{m}S},$$
where *R*_m_ was membrane resistance, and *l* and *S* were the thickness and working area of the membrane, correspondingly.

The diffusion permeability was measured in a two-compartment cell by diffusion of electrolyte solution through the membrane to distilled water. The solution in both chambers was intensively stirred to eliminate the influence of diffusion boundary layers. The changes in solution resistance in the chamber filled with distilled water was used to determine the kinetic increase in solution concentration.

The diffusion flux through the membrane (*j*_m_) and integral coefficient of membrane diffusion permeability (*P*_m_) were calculated by the formulas
(2)$$j_{m}=\frac{V}{S}K\frac{d1/R}{dt},$$
(3)$$P_{m}=\frac{j_{m}l}{C},$$
where *C* was the electrolyte concentration, *V* was the chamber volume, *K* was the cell constant, *R* was solution resistance in the chamber filled with distilled water, and *t* was the time. 

The current-voltage curves (CV-curves) were measured in cylindrical four-chamber cells, which contained two electrode and two membrane chambers. The electrode chambers were supplied with platinum polarizing electrodes with 7.1 cm^2^ of working area. The membrane chambers were supplied with Lugin-Gabber capillaries held to the membrane surface. The membrane chambers were separated from the platinum polarizing electrodes by auxiliary cation and anion-exchange membranes from the cathode and anode side correspondingly to prevent the intrusion of electrolysis products from electrode to membrane chambers. The chambers were fed with 0.05 M NaCl from the separate tanks; the solution volume velocity in the chambers was 14 mL/min. The constant current was supplied to the platinum polarizing electrodes with the linear sweep 0.1 mA/s by a KEITHLEY 2420 high voltage current source. The potential drop on the membrane was measured by silver-silver chloride electrodes by Lugin-Gabber capillaries connected to with KEITHLEY 2701 multimeter.

### 2.4. Mechanical Properties

The dynamic mechanical analysis of air-dry samples was performed by a tension testing machine with an extension rate of 5 mm/min at room temperature and humidity. The working width and length of the sample were 4.2 and 10 mm.

## 3. Results and Discussion

The equilibrium (IEC, water content, specific water content) and transport (conductivity, diffusion permeability, CV-curves) properties for experimental membrane samples are investigated in the present work. As can be seen from the data presented in Table 1, the IEC and water content linearly decrease with growth of the F-26 portion. Overall, the obtained samples M10–M40 have low water content in comparison with other Nafion-type samples [26]. Lowering in specific water content from 14 to 10 mol water molecules per fixed group points to mitigation the membrane swelling caused by hydrophobicity of F-26 polymeric chains. This effect becomes significant when F-26 content exceeds 20%.

### 3.1. Diffusion Permeability

The concentration dependencies of membrane diffusion permeability in NaCl solutions are presented in Figure 1. As one can see, the 10% addition of F-26 does not affect the diffusion permeability, and the dependencies for M0 and M10 samples are similar. The further increase in F-26 content leads to the reduction of the diffusion flux and permeability of the membrane. Diffusion transport is mainly realized through free solution inside the membrane. Therefore, the reason of this effect is a reduction of the membranes’ water content. A decrease in diffusion permeability is the positive effect of application of the samples in redox-flow batteries, since it indicates the reduction of crossover of charged and uncharged electrochemical particles through the membrane.

### 3.2. Conductivity

Figure 2 presents the concentration dependencies of membrane conductivity in NaCl model solutions. The conductivity of the M0 sample is close to the conductivity of commercial MF-4SK and Nafion membranes obtained by extrusion [26]. As can be seen from the figure, with an increase in the content of the inert component in the membrane, its electrical conductivity decreases and conductivity of M40 is 6 times lower than of M0. The conductivity reduction could be caused by lowering the values of IEC and the water content of the samples. According to common concept, all water inside the membrane could be divided on bonded and free [27]. The increasing shape of conductivity concentration dependence is conditioned by the presence of inner free solution. All concentration dependencies have increasing shapes except for the M40 sample, in which conductivity is independent of solution concentration. This points to a small amount of internal equilibrium solution inside the M40 membrane and corresponds to its low water content.

### 3.3. Parameters of Extended Three-Wire Model

The concentration conductivity curves are used for calculation of the transport-structural parameters of the extended three-wire model (ETWM) for obtained membranes. This model is based on the microheterogeneous structure of the ion-exchange material and the theory of generalized conductivity of structurally heterogeneous media. According to the model, the ion-exchange material consists of a gel phase, which includes all polymer components and bond water, and an intergel phase that is an inner equilibrium solution. In the gel phase, the current is transferred by counter-ions, but in the intergel phase it is transferred by both counter- and co-ions. The resulting conductivity of such a two-phase system could be calculated by the formula [28]
(4)$$\kappa_{m}={\left[f_{1}\kappa_{\mathrm{iso}}^{\alpha }+f_{2}\kappa_{\mathrm{sol}}^{\alpha }\right]}^{1/\alpha },$$
where *f*_1_, *f*_2_ are volume fractions of gel and intergel phase, correspondingly, *f*_1_ + *f*_2_ = 1; α is the structural parameter reflecting the spatial orientation of conducting phases inside the material: α = 1 corresponds to the parallel orientation of conducting phases towards the transport direction, α = −1–to the serial one; and κ_m_, κ_iso_, κ_sol_ are conductivities of membrane, its gel phase and equilibrium solution, respectively. According to ETWM, the current is transferred by tree parallel channels: successively through gel phase and solution (*a* in Figure 3), only through gel phase (*b* in Figure 3) and only through solution (*c* in Figure 3). Current fractions, passing through every channel, are described by parameters *a*, *b* and *c* (*a* + *b* + *c* =1), parameters *d* and *e* correspond to fractions of solution and gel in channel *a* (*d* + *e* = 1).

The main equations of ETWM combines the transport (κ_iso_), structural (*f*_1_, *f*_2_, α) and geometrical parameters (*a*, *b*, *c*, *d*, *e*)
(5)$$K_{m}=\frac{aK_{d}}{e+dK_{d}}+bK_{d}+c,$$
(6)$$b=f_{1}{}^{1/\alpha },$$
(7)$$c=f_{2}{}^{1/\alpha },$$
(8)$$a=1-f_{2}^{1/\alpha }-f_{1}^{1/\alpha },$$
(9)$$d=1-(f_{1}-b)/a,$$
(10)$$e=\left(f_{1}-b\right)/a,$$
where *K*_m_ and *K*_d_ are conductivities of membrane and its gel phase divided to solution conductivity, correspondingly, $K_{m}=\frac{\kappa_{m}}{\kappa_{\mathrm{sol}}}$, $K_{d}=\frac{\kappa_{\mathrm{iso}}}{\kappa_{\mathrm{sol}}}$. On the assumption that current in a and b channels is transferred by counter-ions, and in channel c—by both co- and counter-ions, the counter-ions transport number (${\bar{t}}_{+}$) could be calculated by the formula
(11)$${\bar{t}}_{+}=1-t_{-}\frac{c}{K_{m}},$$
where *t*_–_ is co-ion transport number in electrolyte solution. This assumption is adequate in diluted solutions with concentration *C* < 1 mol/L, where the Donnan electrolyte sorption could be neglected. The ETWM permits the determination of structural (*f*_1_, *f*_2_, α) and geometrical (*a*, *b*, *c*, *d*, *e*) parameters simultaneously, which discovers a wide opportunity in discussion of ion-exchange membranes conductivity. Thus, this model is used for the simplified parametrization of the ion-exchange membrane.

Figure 4 presents the calculated ETWM parameters dependencies on a portion of F-26 (ω) in membrane. As can be seen, the value of the gel phase fraction in membrane steadily increases as the inert polymer portion grows and achieves its limiting value *f*_1_ = 1 for the M40 sample. This correlates with a decrease in the fraction of inner free solution due to lower swelling of the membranes. The conductivities of the membrane phase (κ_iso_) also decreases with IEC, but other α, *a*, *b*, *c*, *d* and *e* parameters vary stepwise. These parameters are almost constant in the F-26 content range from 0 to 30%, but for the M40 sample they change considerably due to the reorganization of its structure. Thus, the current transfer through mixed and solution channels (*a* and *c* parameters) decreases. The value of α parameter is 2 times greater for the M40 membrane than for other samples that point to the domination of parallel orientation of conducting phases, while for other samples the orientation of conducting phases is more chaotic. The similar effect of stepwise change in model parameters was observed earlier for heterogeneous cation exchange membranes with variable content of inert polyethylene binders when the binder content was 40% [29].

The membrane in RFB plays an important role in transporting the counter-ions from one chamber to another, and to retard the co-ions flux. In contrast to the PEMFC, where there is only one type of cations and water, in RFB the membrane contacts the solutions, containing both counter- and co-ions, so the selectivity of the membrane is important. The calculated transport-structural parameters are used to estimate the counter-ion transport numbers ${\bar{t}}_{+}$ by the Formula (11). The concentration dependencies of ${\bar{t}}_{+}$ are presented in Figure 5. One can see that samples with inert polymer content in the range 0–20% have close values of ${\bar{t}}_{+}$, and an increase in F-26 content up to 30% leads to the significant reduction of membrane selectivity due to its low IEC. The M40 sample has a higher volume fraction of gel phase and a lower value of c parameter so the M40 sample has a higher transport number than the M30 one. Nevertheless, all the investigated membranes have high selectivity, and counter-ion transport numbers are larger than 0.975 in 1 M NaCl solution.

Thus, on the base of model parameter analysis it could be concluded that the introduction of 20% inert polymer into perfluorinated membrane does not affect the structure and selectivity of the membrane.

### 3.4. Microheterogeneous Model

The concentration dependencies of conductivity and diffusion permeability (Figure 1 and Figure 2) are processed within the two-phase microheterogeneous model (MM) to estimate transport-structural parameters of the model. According to the model, the material is also presented as a microheterogeneous medium with the phase grouping similar to ETWM. The model is described in detail in [25,28]. This model includes *f*_1_, *f*_2_ and κ_iso_, which could be calculated from the bi-logarithmic presentation of membrane conductivity dependence on solution conductivity according to formula
(12)$$\kappa_{m}=\kappa_{\mathrm{iso}}^{f_{1}}\kappa_{\mathrm{sol}}^{f_{2}},$$

The MM also permits to calculate parameters *β* and *G*, characterizing the shape of concentration profile inside the membrane and the diffusion transport of co-ions in the membrane gel phase. These parameters *β*, *α* and *G* could be calculated according to the formulas
(13)$$\beta =\frac{\ln j}{\ln C},$$
(14)$$\alpha =\frac{\ln[f_{2}/(2-\beta )]}{\ln\left(P_{m}^{*}/D\right)},$$
(15)$$G=\frac{P_{m}^{*}}{C}{\left(\frac{\left(\beta -1\right)}{f_{1}}\right)}^{1/\alpha },$$
where $P_{m}^{*}$ is the differential coefficient of diffusion permeability, and $P_{m}^{*}=P_{m}\cdot \beta$; *D* is the diffusion coefficient of the electrolyte in solution.

Table 2 shows the transport-structural parameters of MM. As can be seen, values of *f*_2_, κ_iso_ and *α* are close to the calculated ones according to ETWM.

This fact points to the agreement of the two models’ results. It is worth noting the parameters *β* and *G*, which characterize the diffusion transfer in membranes. Thus, the value of the *β* parameter, corresponding to the shape of concentration profile inside the membrane, is almost independent on the F-26 portion for samples M0, M10, and M20, while the inert polymer portion exceeds 20% of the decrease in *β* value,, and it equals 1 for the M40 membrane due to the significant decrease in the IEC of the sample. The *G* parameter includes the Donnan constant, the IEC of the gel phase and the diffusion coefficient of co-ions in the gel phase [30] and describes the diffusion of co-ions in the membrane gel phase. The co-ion transfer also depends on the proportion of the internal equilibrium solution in the membrane. The *G* parameter magnitude also decreases with the growth of F-26 content inside the M0–M30 membranes. The manifestation of the effect of obstacles due to the introduction of an inert polymer is also not excluded. For the M40 sample this parameter could not be calculated due to the model restrictions.

### 3.5. CV-Curves

During the employment the membranes in effective current generation regimes the diffusion limited current on the membrane could be achieved. Therefore, the main purpose of CV-curve measurement was to control the changes in limiting the current density and Ohmic slope after modification. The CV-curves for samples with different portions of inert polymers are experimentally investigated. The investigated membranes are prepared by the casting method, so the membrane surfaces are faced to glass and air during drying. In [31] the authors observed the asymmetry of CV-curves for the membranes prepared by the casting method. In the present work, we also measure the CV-curve for M0 sample for its different orientation to the electrolyte flux to reveal the asymmetry effect [32]. The CV-curves for both orientations of the membrane are similar, so we decided to consider the other samples also being symmetric.

The CV-curves for investigated membranes are presented in Figure 6, Table 3 shows the parameters of CV-curves. As can be seen, the limiting current density (*i*_lim_) and potential of transition to limiting state (Δ*E*_lim_), as well as slopes of Ohmic (tg_ohm_) and overlimiting (tg_over_) sections are almost equal within the experimental error. Some change in slope of limiting plateau indicates the uneven achievement of limiting state on the membrane surface.

The analysis of limiting plateau length (Δ) has shown that its dependence on inert polymer portion inside the membrane has minimum for the M20 sample. The introduction of inert polymer into the homogeneous one leads to an increase in electric inhomogeneity of the membrane surface and the intensification of electroconvection, so the overlimiting state is achieved for lower value of potential drop on the membrane [33]. The highest value of limiting plateau slope (tg_lim_) and lowest length of limiting plateau (Δ, V) are also observed for M20 membrane. This could indicate the highest development of electroconvection in this sample due to the optimal ratio of conductive and non-conductive areas on the membrane surface among the studied samples.

### 3.6. Mechanical Properties

The dynamometric curves for membranes are shown in Figure 7, which are used for calculation of the mechanical characteristics of the membranes: Young’s modulus (*E*), yield strength, tensile strength and relative elongation (ε). The dynamometric parameters are presented in Table 4.

One can see, that the increase in F-26 content in the membrane is accompanied by the reduction of Young’s modulus and tensile strength values by 30% and yield strength by 50%. At the same time, the relative elongation increases by a factor of three times. It should be noted that elongation dependence on the fraction of inert polymer is linear for M0–M40 membranes, so F-26 acts as plasticizer in the obtained polymer mixture and facilitates the mutual slide of polymer chains. Young’s modulus, yield strength, tensile strength and relative elongation values dramatically decrease with the growth of membrane water uptake [34,35] due to formation of pores and cavities which act as microcracks and significantly reduce the strength of the polymer. Two opposite factors influence the mechanical characteristics of the investigated membranes: a reduction in water content that strengthens the membrane, and an increase in F-26 content with lower mechanical properties.

The remaining characteristics change nonmonotonically: they decrease in the range of ω from 0 to 20%, but for samples M20 and M40 they have close values. The growth of membrane elasticity is a positive effect for their application in electricity chemical sources, since formation of the membrane electrode assembly for both fuel cells and redox flow batteries is performed by the pressing of electrodes to the membrane surface. Thus, higher elasticity of the membrane could provide the better contact between electrodes and membranes in the stack. According to [7], the mechanical characteristics are closely related to crossover in the direct methanol fuel cell due to the formation of cracks and pinholes under DMFC operation conditions. Therefore, the membrane with high flexibility and low rigidity is preferred to sustain mechanical stresses and prevent the membrane from breaking or perforating during DMFC operating [7,36].

## 4. Conclusions

A series of composite membranes with varying content of inert component from 0 to 40 wt.% on a dry membrane was obtained by casting from solutions of perfluorosulfonic acid and poly(vinylidene fluoride co-hexafluoropropylene) in dimethylformamide. It is shown that with an increase in the mass fraction of an inert fluoropolymer, the ion-exchange capacity and hydration characteristics of membranes naturally decrease. A simultaneous decrease in the diffusion permeability of membranes for a sodium chloride solution indicates a possible decrease in the crossover of charged and uncharged electrochemical particles through the membrane during its operation in a fuel cell or redox flow battery.

The concentration dependence of the specific electrical conductivity of membranes in sodium chloride solutions was studied by the mercury-contact method, and a decrease in electrical conductivity by a factor of six with an increase in the proportion of the inert component from 0 to 40% was shown. In this case, the introduction of an inert fluoropolymer into the membrane has practically no effect on the limiting diffusion current and other parameters of the current-voltage curve.

Within the framework of the extended three-wire model of membrane conductivity, the parameters characterizing the current paths in the membrane were determined which permitted the estimation of the sodium counter-ion transport numbers in the membranes. It was is shown that a significant change in the structure of membranes is observed when an inert polymer exceeds 30%. In this case, the paths of current flow through the membrane are reorganized, and its selectivity decreases.

Based on the dynamometric analysis, the effect of an inert fluoropolymer on the mechanical characteristics of perfluorinated membranes was determined. With an increase in the proportion of the inert component from 0 to 40%, a decrease in the value of Young’s modulus and tensile strength by about 30% and yield strength by 50% was found. In this case, the relative elongation linearly increases by a factor of three, which indicates the plasticizing effect of the inert fluoropolymer.

Summarizing the information on the effect of the proportion of the inert component on the physicochemical and electrotransport properties of perfluorinated membranes, it can be concluded that the optimal amount of the inert polymer F-26 in the perfluorinated membrane is 20%. This does not significantly affect the structure and electrotransport properties of the membrane, but increases its elasticity and reduces diffusion permeability, which is important when such materials are used in electric current generation devices.

## Figures and Tables

**Figure 1 membranes-12-00935-f001:**
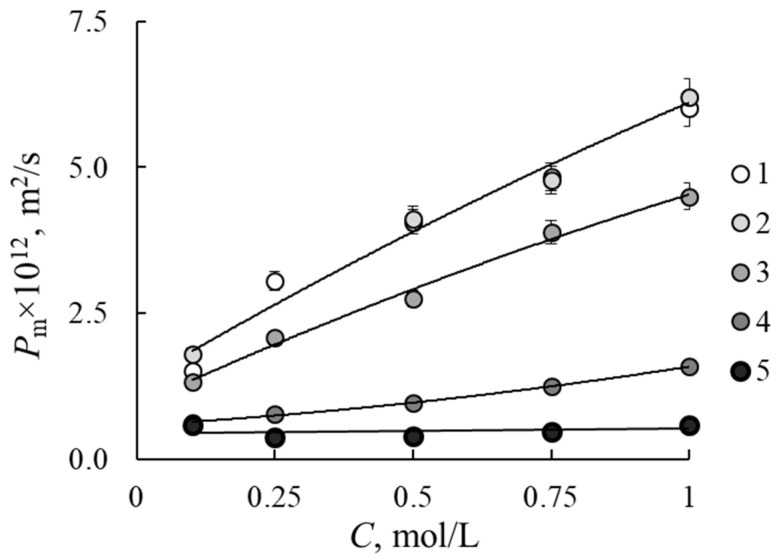
Concentration dependencies of membranes diffusion permeability in NaCl solutions: 1—M0, 2—M10, 3—M20, 4—M30, 5—M40.

**Figure 2 membranes-12-00935-f002:**
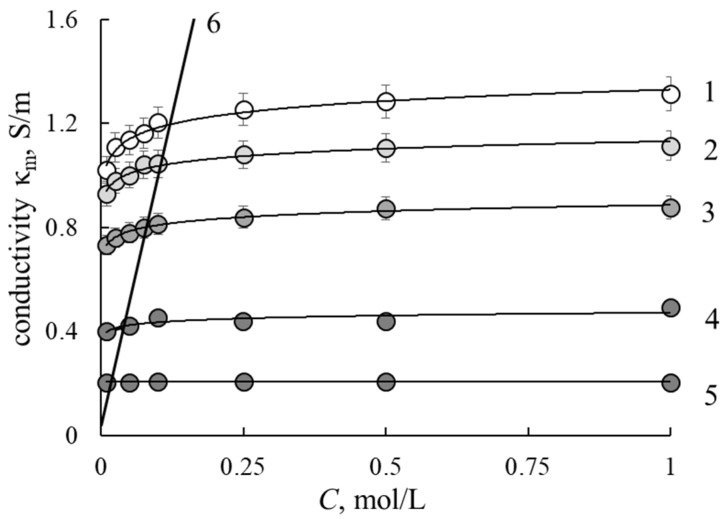
Conductivity concentration dependencies for investigated membranes with different portions of inert polymer: 1—M0, 2—M10, 3—M20, 4—M30, 5—M40, 6—NaCl solution.

**Figure 3 membranes-12-00935-f003:**
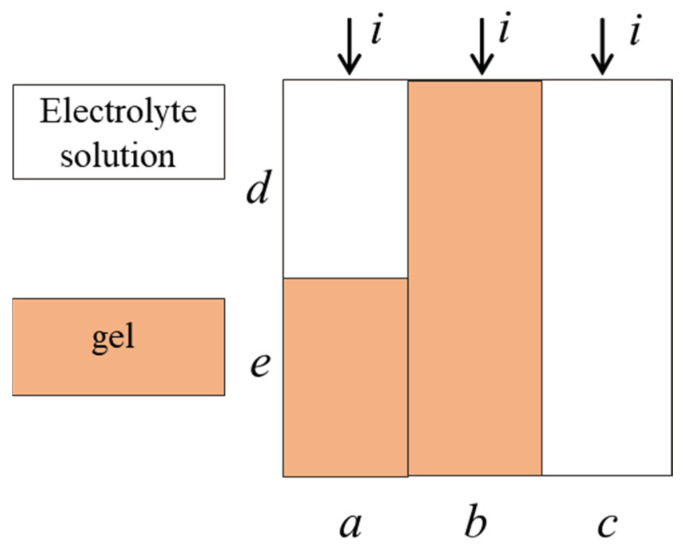
Scheme of current transfer in ion-exchange material in within ETWM. *a*, *b*, *c* are fractions of current passing through mixed, gel and solution channels (*a* + *b* + *c* =1); *d*, *e* are fractions of solution and gel in mixed channel *a* (*d* + *e* = 1).

**Figure 4 membranes-12-00935-f004:**
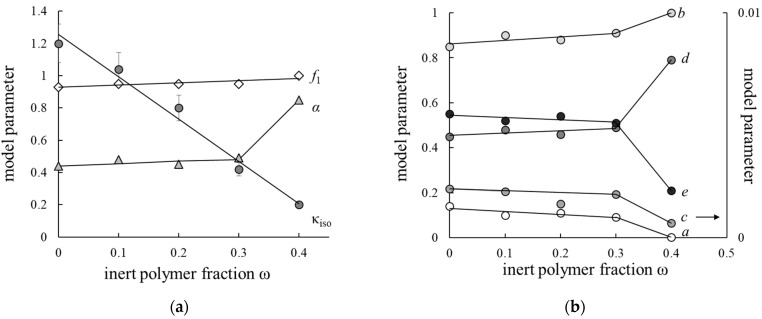
Values of transport-structural (**a**) and geometric (**b**) parameters of ETWM as dependencies on F-26 content in membranes. *f*_1_ is volume fractions of gel phase; α is the structural parameter reflecting the spatial orientation of conducting phases inside the material; *a*, *b*, *c* are fractions of current passing through mixed, gel and solution channels (*a* + *b* + *c* =1); *d*, *e* are fractions of solution and gel in mixed channel *a* (*d* + *e* = 1).

**Figure 5 membranes-12-00935-f005:**
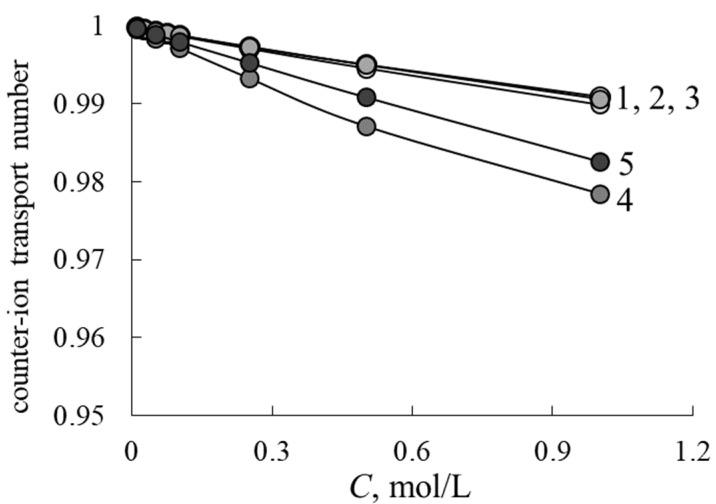
Concentration dependencies of counter-ion transport numbers in NaCl solutions: M0 (1), M10 (2), M20 (3), M30 (4), M40 (5).

**Figure 6 membranes-12-00935-f006:**
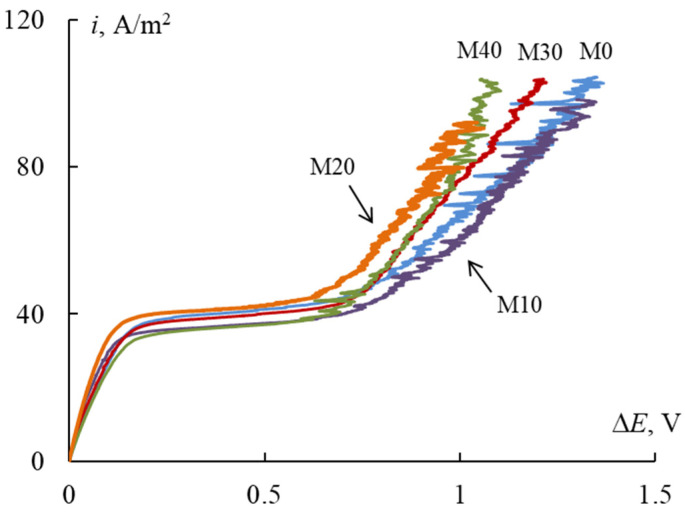
CV-curves for investigated membranes.

**Figure 7 membranes-12-00935-f007:**
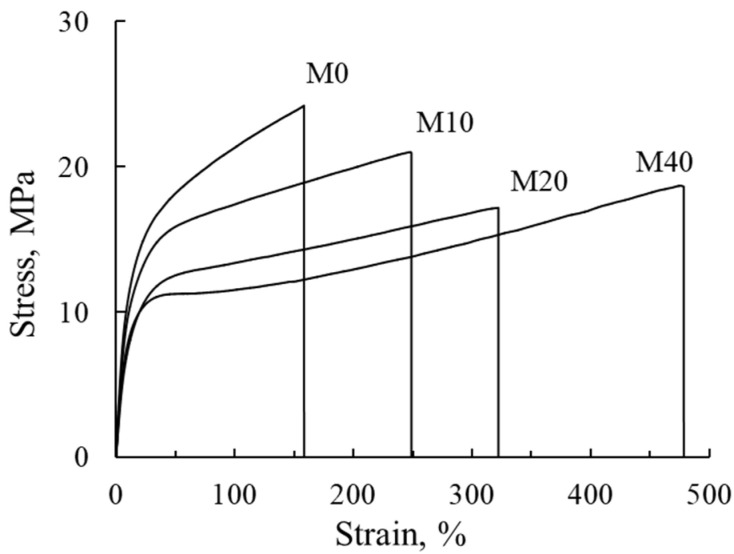
Engineering stress–strain curves of the membranes: M0–M40.

**Table 1 membranes-12-00935-t001:** Physicochemical characteristics of the membrane samples in Na^+^-form.

Membrane	ω, %	l, μm	Q, mmol/g_dry_	W, %	n_m_, molH_2_O/molSO_3_^−^
M0	0	359 ± 20	1.02	21.3	14
M10	10	398 ± 20	0.91 ± 0.01	17.8	13
M20	20	437 ± 20	0.80 ± 0.01	16.1	13
M30	30	368 ± 20	0.67 ± 0.01	12.9	12
M40	40	367 ± 20	0.60 ± 0.01	9.9	10

**Table 2 membranes-12-00935-t002:** Transport-structural parameters of MM.

Membrane	*β*	*f* _2_	κ_iso_, S/m	*α*	G∙10^15^, m^5^ mol ^−1^ s^−1^
M0	1.54	0.06	1.19	0.38	2.89
M10	1.50	0.04	1.03	0.45	2.94
M20	1.54	0.04	0.80	0.40	2.06
M30	1.35	0.04	0.42	0.40	0.20
M40	1.00	0.0002	0.21	1	-

**Table 3 membranes-12-00935-t003:** Parameters of CV-curves for investigated membranes.

Membrane	*i*_lim_, A/m^2^	ΔE_lim_, V	ΔE_over_, V	Δ, V	tg_ohm_	tg_lim_	tg_over_
M0	37.6 ± 1.6	0.13 ± 0.004	0.88 ± 0.13	0.84 ± 0.14	283.06 ± 5.29	8.04 ± 0.41	127.82 ± 27.84
M10	34.8 ± 0.7	0.09 ± 0.001	0.81 ± 0.03	0.72 ± 0.03	369.94 ± 6.37	7.34 ± 0.14	115.88 ± 8.41
M20	38.2 ±0.5	0.11 ± 0.003	0.63 ± 0.03	0.52 ± 0.03	326.68 ± 6.44	11.39 ± 1.7	126.12 ± 14.11
M30	36.9 ± 0.1	0.12 ± 0.001	0.71 ± 0.01	0.59 ± 0.01	317.46 ± 6.45	7.74 ± 0.22	120.20 ± 1.98
M40	36.4 ± 0.9	0.11 ± 0.001	0.80 ± 0.07	0.72 ± 0.07	299.39 ± 0.85	7.71 ± 4.35	121.30 ± 27.26

**Table 4 membranes-12-00935-t004:** Dynamometric characteristics of the membranes.

Membrane	*E*, MPa	Yield Strength, MPa	ε, %	Tensile Strength, MPa
M0	176	16.7	160	24.2
M10	142	15.4	250	20.9
M20	108	12.1	320	17.2
M40	135	11.0	480	18.7
F-26	–	–	520	40.9

## Data Availability

Not applicable.

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
