# Peer review of "Nanocomposite Membranes Based on Fluoropolymers for Electrochemical Energy Sources"

_membranes, 2022, doi:10.3390/membranes12100935_

Round 1

Reviewer 1 Report

The manuscript “Nanocomposite Membranes Based on Fluoropolymers for Electrochemical Energy Sources” is aimed at the physicochemical and transport properties analysis: exchange capacity, water content, diffusion permeability, conductivity, and current-voltage characteristic of a series of perfluorinated membranes.  The mechanical characteristics of the membranes, such as Young's modulus, yield strength, tensile strength, and relative elongation, were also estimated from the dynamometric curves.

 The mechanical properties of the membranes are measured thoroughly, and the results are interpreted clearly. The electrochemical measurements are presented mainly by the CV-curves for investigated membranes. One can see the limiting current density and potential of transition to limiting state for all membranes under study are closely similar within the experimental error.

 Methodology of the paper is well performed, the structure of the paper is valid, experimental part clarifies the conclusions. The overall conclusion about the optimal amount of the inert polymer F-26 in the perfluorinated membrane seems to be valid and of interest. It was found to be 20%, which does not significantly affect its structure and electrotransport properties, but increases the elasticity of the obtained samples.

 I will highly appreciate if the Introduction part of the manuscript would cite the following works, that I think are on the subject. To emphasize the importance of the membrane materials not only on redox flow batteries and fuel cells, but on the hybrid systems I recommend citing Dr. Modestov, Dr. Tang and Dr. Kim works:

A Hydrogen–Bromate Flow Battery for Air-Deficient Environments

https://doi.org/10.1002/ente.201700447

·       Redox flow batteries: importance in modern electrical energy industry and comparative characteristics of the main types

http://dx.doi.org/10.1070/RCR4987

·       Progress and Perspective of the Cathode Materials towards Bromine-Based Flow Batteries

https://doi.org/10.34133/2022/9850712

·       Ultrathin Nafion-Filled Porous Membrane for Zinc/Bromine Redox Flow Batteries

https://doi.org/10.1038/s41598-017-10850-9

Author Response

We are grateful to the Reviewer for their time and efforts aimed at improving our manuscript. We find recommendations quite useful; all references are added to the text (Ref. 2, 4, 11, 19). They improved the review in Introduction section. The corrections are marked up using the “Track Changes” function.

With my best regards,

Irina Falina

Reviewer 2 Report

The Authors used the Nanocomposite Membranes Based on Fluoropolymers for Electrochemical Energy Sources. the manuscript is written well and did not use advanced equipment to prove them idea. 

1- the whole manuscript should check the grammar.

2- they should used another tools to confirm the active materials.

3- the authors should prepared table to comaper between the current result and pervious result

Author Response

We are grateful to the Reviewer for their time and efforts aimed at improving our manuscript. We find your comments and recommendations quite useful. We have revised our manuscript according to your recommendations. The corrections are marked up using the “Track Changes” function. The Reviewer’s comments and our responses are presented below.

With my best regards,

Irina Falina

Reviewer’s comments

The Authors used the Nanocomposite Membranes Based on Fluoropolymers for Electrochemical Energy Sources. the manuscript is written well and did not use advanced equipment to prove them idea. 

Comment 1: the whole manuscript should check the grammar.

Reply: We thoroughly revised all the text and checked the spelling. All corrections are introduced to the text.

Comment 2: they should used another tools to confirm the active materials.

Reply: The tests with Fenton reagent are typical for membranes, which are proposed for application in electrochemical energy sources. We can also provide the experiments with Fenton reagent for our samples, if the editor would be so kind to provide us an extra time.

Comment 3: - the authors should prepared table to comaper between the current result and pervious result

Reply: Thank you for your comment. The aim of the current work was only to investigate the influence of inert polymer content on characteristics of Nafion-type membrane. Therefore, we prepared thick samples that could be compared with commercial samples having similar thickness or with each other. In our future work we are planning to prepare the more appropriate samples for testing in RFB of DMFC single cells.

Reviewer 3 Report

The manuscript reported the physicochemical and transport properties of a series of perfluorinated membranes with an inert fluoropolymer content, obtained by polymer solution casting. The optimal amount of the inert polymer in the perfluorinated membrane was found to be 20%, which does not significantly affect its structure and electron transport properties but increases the elasticity. Hence, it is promising for application in redox flow batteries and direct methanol fuel cells.

I consider the content of this manuscript will definitely meet the reading interests of the readers of the Membranes journal. However, there are certain English spelling and grammar issues, and also the discussion and explanation should be further improved. Therefore, I suggest giving a minor revision and the authors need to clarify some issues or supply some more experimental data to enrich the content.

1. For grammar issues, it is suggested that the author double-check the small grammar errors in the full text, especially the lack of and redundant use of definite articles.

  Just for the abstract part:

  Line 11, ‘obtained by polymer solution casting from, were studied’. ‘from’ should be deleted.

  Line 18, ‘Therefore, the perfluorinated membrane with 20 % is promising for application in redox flow batteries and direct methanol fuel cells.’ It is not so clear with 20% of what, and it should be clarified.

2. For the Keywords, ‘flow battery’, ‘direct methanol fuel cell’, and ‘solution casting’ should be added in order to attract a broader readership.

3. Line 24, ‘The expansion of alternative energy applications is closely related to the development of redox flow batteries (RFB)’. But why? This is not explained very well at the beginning. Most renewable energy sources are intermittent, opening spatial and temporal gaps between the availability of the energy and its consumption by the end-users. In order to address these issues, it is necessary to develop suitable energy storage systems for the power grid, such as RFBs [Electrochimica Acta 309 (2019): 311-325].

4. It is not only just to achieve high Coulomb efficiency (Line 28), as for energy storage, energy efficiency is the most important. Imagine a membrane serves as an efficient barrier layer for active species and with low crossover, so very high Coulomb efficiency but at the same time if the resistance is also high, the VE will be very low [Electrochimica Acta 378 (2021): 138133]. Finally, this kind of membrane will demonstrate very low energy efficiency and is not suitable for RFBs.

5. Line 63, ‘So, during operation of both battery, and hydrogen fuel cell, peroxide radicals are generated [17].’ Is it possible for RFBs to generate peroxide radicals during operation?

6. Line 84, ‘F-26 solution in DMF was prepared by ultrasound dissolution of F-26 powder in DMF.’ This may confuse the readers, about whether it is an F-26 solution or powder dissolved in DMF. It should be clarified, that maybe the F-26 solution is dried first to remove the solvent and F-26 powder is obtained. Then the powder is dissolved in DMF.

7. In Table 1, it is clear that the thickness of M20 differs from the others. What is the reason and why this thickness cannot be better controlled? The influence of the membrane on the performance has been reported in VRFB, so this thickness difference may make the comparison not very reasonable [Journal of Membrane Science, 2016, 510, pp. 18-26; International Journal of Energy Research 2019, 43(14), pp. 8739-8752]. Line 139, ‘This effect becomes significant when F-26 content exceeds 20 %’. I suspect that this effect becomes significant due to the sudden thickness increase.

8. How about the chemical stability and thermal stability of the membrane? For VRFB applications, chemical stability is mainly conducted in V5+ solutions; While for DMFC, the Fenton reagents are typically used for chemical stability. Thermal stability by TGA and DSC may also be very useful and important for DMFC which does not operate at room temperature, but at elevated temperatures [International Journal of Hydrogen Energy 2020, 45(13), pp. 7829-7837].

9. Line 219, ‘Figure 4. Values of transport structural and (a) and geometric (b) parameters of ETWM as dependencies on F-26 content in membranes.’

10. Line 269, ‘In present work, we also measure the CV-curve for M0 sample for its different orientation to the electrolyte flux to reveal the asymmetry effect [27].’ Normally, the CV test of electrodes in VRFB solutions can be carried out to see the reversibility of the electrodes. But what is the purpose of doing a CV-curve for membranes? It is still not very clear, because no redox reactions take place on the surface of the membrane, but on the surface of the electrode. I suggest here explaining a bit more about the purposes for the CV-curve of membranes.

11. Line 313, ‘The growth of membranes elasticity is a positive effect for their application in electricity chemical sources since formation of a membrane electrode assembly for both fuel cells and redox flow  batteries is performed by pressing of electrodes to the membrane surface.’ It is clear that the obtained membranes are designed for RFB and DMFC applications. But why no DMFC and RFB single-cell tests are carried out? It makes more sense to combine the electrochemical performance of the related single-cells assembled with the prepared polymer-blending membranes with the above-mentioned Physico-chemical properties of the membranes.

Author Response

We are grateful to the Reviewer for their time and efforts aimed at improving our manuscript. We find your comments and recommendations quite useful. We have revised our manuscript according to your recommendations. The corrections are marked up using the “Track Changes” function. The Reviewer’s comments and our responses are presented below.

With my best regards,

Irina Falina

Reviewer’s comments

I consider the content of this manuscript will definitely meet the reading interests of the readers of the Membranes journal. However, there are certain English spelling and grammar issues, and also the discussion and explanation should be further improved. Therefore, I suggest giving a minor revision and the authors need to clarify some issues or supply some more experimental data to enrich the content.

Comment 1. For grammar issues, it is suggested that the author double-check the small grammar errors in the full text, especially the lack of and redundant use of definite articles.

  Just for the abstract part:

  Line 11, ‘obtained by polymer solution casting from, were studied’. ‘from’ should be deleted.

  Line 18, ‘Therefore, the perfluorinated membrane with 20 % is promising for application in redox flow batteries and direct methanol fuel cells.’ It is not so clear with 20% of what, and it should be clarified.

We carefully checked the spelling and thoroughly revised all the text.

Comment 2. For the Keywords, ‘flow battery’, ‘direct methanol fuel cell’, and ‘solution casting’ should be added in order to attract a broader readership.

Reply: The Keywords are changed as follows

Keywords: perfluorinated membrane; inert fluoropolymer; conductivity; diffusion permeability; current-voltage curve; stress-strain curve; extended tree-wire model; redox flow battery; direct methanol fuel cell; solution casting.

Comment 3. Line 24, ‘The expansion of alternative energy applications is closely related to the development of redox flow batteries (RFB)’. But why? This is not explained very well at the beginning. Most renewable energy sources are intermittent, opening spatial and temporal gaps between the availability of the energy and its consumption by the end-users. In order to address these issues, it is necessary to develop suitable energy storage systems for the power grid, such as RFBs [Electrochimica Acta 309 (2019): 311-325].

Reply: Thank you for the comment. The sentence is changed as follows

The expansion of alternative energy applications is closely related to the development of redox flow batteries (RFB) with aqueous and non-aqueous electrolytes, which are widely recognized easy for scale up and suitable for large-scale energy storage applications (10 kW–10 MW). The total energy output depends on the volume of the reservoirs [1, 2].

Comment 4. It is not only just to achieve high Coulomb efficiency (Line 28), as for energy storage, energy efficiency is the most important. Imagine a membrane serves as an efficient barrier layer for active species and with low crossover, so very high Coulomb efficiency but at the same time if the resistance is also high, the VE will be very low [Electrochimica Acta 378 (2021): 138133]. Finally, this kind of membrane will demonstrate very low energy efficiency and is not suitable for RFBs.

Reply: Thank you for the comment. The sentence is changed as follows

There are a number of requirements for RFB membranes, including high conductivity, thermal and chemical stability, mechanical strength, low permeability to electrochemically active components to achieve high Coulomb and energy efficiency.

We should note that in many cases, the membrane resistance is just the part of total cell resistance, and resistances of both solution and interface electrode/electrolyte also considerably contribute to the total resistance of the system. The reference is added to the text (Ref. 5).

Comment 5. Line 63, ‘So, during operation of both battery, and hydrogen fuel cell, peroxide radicals are generated [17].’ Is it possible for RFBs to generate peroxide radicals during operation?

Reply: The sentence is corrected as follows

So, during operation of both battery, and hydrogen fuel cell, membrane is subjected to highly active oxidants.

Comment 6. Line 84, ‘F-26 solution in DMF was prepared by ultrasound dissolution of F-26 powder in DMF.’ This may confuse the readers, about whether it is an F-26 solution or powder dissolved in DMF. It should be clarified, that maybe the F-26 solution is dried first to remove the solvent and F-26 powder is obtained. Then the powder is dissolved in DMF.

Reply: The sentence is corrected as follows

A solution of F-26 in DMF was prepared by ultrasonic dissolution of F-26 powder in DMF.

We hope, it has become clearer.

Comment 7. In Table 1, it is clear that the thickness of M20 differs from the others. What is the reason and why this thickness cannot be better controlled? The influence of the membrane on the performance has been reported in VRFB, so this thickness difference may make the comparison not very reasonable [Journal of Membrane Science, 2016, 510, pp. 18-26; International Journal of Energy Research 2019, 43(14), pp. 8739-8752]. Line 139, ‘This effect becomes significant when F-26 content exceeds 20 %’. I suspect that this effect becomes significant due to the sudden thickness increase.

Reply: We agree with reviewer, and membrane thickness effects on its resistance. So, the thicker membrane is, the higher resistance it has, and lower energy efficiency is obtained from the RFB. The aim of the present work is to investigate the influence of inert polymer content on characteristics of the membranes. The objects of research were laboratory samples. Most of investigated characteristics account the membrane thickness during the data treatment. So, we considered, that higher thickness of M20 sample is insignificant. Line 129 discusses the IEC, water uptake and specific water content of the membranes. All these parameters are independent on the membrane thickness, as far as pore water distribution and internal surface area.

Comment 8. How about the chemical stability and thermal stability of the membrane? For VRFB applications, chemical stability is mainly conducted in V5+ solutions; While for DMFC, the Fenton reagents are typically used for chemical stability. Thermal stability by TGA and DSC may also be very useful and important for DMFC which does not operate at room temperature, but at elevated temperatures [International Journal of Hydrogen Energy 2020, 45(13), pp. 7829-7837].

Reply: The reviewer is right, ant tests with Fenton reagent are typical for membranes, which are proposed for application in electrochemical energy sources. In addition, TGA or DSC are informative for materials, which should be used under evaluated temperature conditions. We can provide the experiments with Fenton reagent for our samples, if the editor would be so kind to provide us an extra time. DSC experiments would be performed for the separate manuscript devoted to structure and water distribution in such materials.

Comment 9. Line 219, ‘Figure 4. Values of transport structural and (a) and geometric (b) parameters of ETWM as dependencies on F-26 content in membranes.’

Reply: The figure capture is corrected.

Values of transport-structural (a) and geometric (b) parameters of ETWM as dependencies on F-26 content in membranes.

Comment 10. Line 269, ‘In present work, we also measure the CV-curve for M0 sample for its different orientation to the electrolyte flux to reveal the asymmetry effect [27].’ Normally, the CV test of electrodes in VRFB solutions can be carried out to see the reversibility of the electrodes. But what is the purpose of doing a CV-curve for membranes? It is still not very clear, because no redox reactions take place on the surface of the membrane, but on the surface of the electrode. I suggest here explaining a bit more about the purposes for the CV-curve of membranes.

Reply: We measured the CV-curves of membrane in a free-standing state, this method is traditionally used for membranes characterization for different applications in electromembrane processes [Tanaka Y. Ion Exchange Membranes. http://dx.doi.org/10.1016/B978-0-444-63319-4.00001-8]. We also expected that modification would effect on the main transport properties of the membrane: slope of Ohmic region and limiting current density. CV-method turned out to be insensitive to the membrane content despite the fact that conductivity of M40 sample is 6 times lover then of M0 one. The next sentences describing the purpose of CVCs measuring are added to the text

During the employment the membranes in effective current generation regimes the diffusion limited current on the membrane could be achieved. Therefore, the main purpose of CV-curves measurement was to control the changes in limiting current density and Ohmic slope after modification.

We also added the next paragraph to section 3.3

The membrane in RFB plays an important role to transport the counter-ions from one chamber to another, and to retard the co-ions flux. In contrast to the PEMFC, where there is only one type of cations and water, in RFB membrane contacts with solutions, containing both counter- and co-ions, so the selectivity of the membrane is important.

Comment 11. Line 313, ‘The growth of membranes elasticity is a positive effect for their application in electricity chemical sources since formation of a membrane electrode assembly for both fuel cells and redox flow  batteries is performed by pressing of electrodes to the membrane surface.’ It is clear that the obtained membranes are designed for RFB and DMFC applications. But why no DMFC and RFB single-cell tests are carried out? It makes more sense to combine the electrochemical performance of the related single-cells assembled with the prepared polymer-blending membranes with the above-mentioned Physico-chemical properties of the membranes.

Reply: Thank you for your comment. We do not have the appropriate equipment for testing DMFC or RFB single sells. But we consider that if we know the requirement to membranes for application in such devices, we can prepare the appropriate membrane.
